# Genetic Risk of Ankylosing Spondylitis and Second-Line Therapy Need in Crohn’s Disease: A Mendelian Randomization Study

**DOI:** 10.3390/jcm13247496

**Published:** 2024-12-10

**Authors:** Mahmud Omar, Mohammad Omar, Yonatan Shneor Patt, Offir Ukashi, Yousra Sharif, Adi Lahat, Christian Phillip Selinger, Kassem Sharif

**Affiliations:** 1Faculty of Medicine, Tel-Aviv University, Tel-Aviv 6997801, Israel; mahmudomar70@gmail.com; 2School of Medicine, V. N. Karazin Kharkiv National University, 61022 Kharkiv, Ukraine; mohammed.nasif8@gmail.com; 3Internal Medicine B, Sheba Medical Centre, Ramat Gan 5262000, Israel; yopatt123@gmail.com; 4Department of Gastroenterology, Sheba Medical Center, Tel-Hashomer 5262000, Israel; offirukashi@gmail.com (O.U.); zokadi@gmail.com (A.L.); 5Department of Gastroenterology, Hadassah Medical Center, Jerusalem 91120, Israel; yusrasharif02@gmail.com; 6Leeds Gastroenterology Institute, Leeds Teaching Hospitals, Leeds LS1 3EX, UK

**Keywords:** Crohn’s disease, ankylosing spondylitis, two-sample Mendelian randomization, genetic predisposition, second-line treatment

## Abstract

**Background:** Crohn’s disease (CD) and Ankylosing Spondylitis (AS) are chronic conditions with overlapping inflammatory pathways. This research investigates the genetic association between AS and the requirement for more aggressive therapeutic interventions in CD, suggesting a likelihood of increased severity in CD progression among individuals diagnosed with AS. **Methods:** This study utilized two-sample Mendelian randomization (TSMR) to analyze GWAS datasets for AS and CD requiring second-line treatment. Instrumental variables were selected based on single-nucleotide polymorphisms of genome-wide significance. Analytical methods included inverse-variance weighted (IVW), MR Egger, and other MR approaches, alongside sensitivity analysis, to validate the findings. **Results:** Our results indicated a significant association between AS genetic predisposition and the increased need for second-line treatments in CD. The IVW method showed an Odds Ratio (OR) of 2.16, and MR Egger provided an OR of 2.71, both were statistically significant. This association persisted even after the exclusion of influential outlier SNP rs2517655, confirming the robustness of our findings. **Conclusions:** This study suggests that genetic factors contributing to AS may influence the progression of CD, potentially necessitating more intensive treatment strategies. These findings underscore the importance of early screening in patients with co-existing AS and CD for tailoring treatment approaches, thus advancing personalized medicine in the management of these complex conditions.

## 1. Introduction

Crohn’s disease (CD) and Ankylosing Spondylitis (AS) are chronic inflammatory conditions that significantly impact patients’ quality of life, often imposing a substantial physical, psychological, and economic burden on affected individuals and healthcare systems [1,2]. Crohn’s disease, a type of inflammatory bowel disease (IBD), is characterized by relapsing and remitting inflammation that can affect any part of the gastrointestinal (GI) tract, though it predominantly involves the ileum and colon [3]. Its clinical presentation is diverse, ranging from abdominal pain and diarrhea to systemic symptoms such as fatigue, weight loss, and malnutrition, which arise due to the chronic inflammatory state and nutrient malabsorption [3,4]. The disease’s natural course varies among individuals, with some experiencing mild symptoms and others progressing to severe complications like fistulas, strictures, or abscesses [3,5].

The management of CD is similarly diverse, tailored to disease severity and location. For mild-to-moderate disease, therapies may include azathioprine, methotrexate, or watchful waiting. However, moderate to severe CD often necessitates advanced treatment modalities, including biologic therapies such as anti-tumor necrosis factor (TNF) agents, anti-integrin agents, JAK inhibitors, interleukin (IL)-12/23 inhibitors, or pure IL-23 inhibitors. These biologic therapies have revolutionized CD management, offering improved rates of clinical and endoscopic remission [5]. Despite these advancements, a subset of patients demonstrates refractory disease or loss of response to biologics over time, necessitating treatment escalation [5,6]. Such therapeutic challenges underscore the importance of understanding the factors that influence disease progression and treatment outcomes in CD.

Ankylosing Spondylitis (AS), a type of spondyloarthritis, predominantly affects the axial skeleton, leading to inflammation of the sacroiliac joints and spine [6]. It is characterized by chronic back pain, stiffness, and reduced spinal mobility, with extra-articular manifestations including uveitis, enthesitis, and, less commonly, bowel inflammation [7,8]. The hallmark of AS is the progression of inflammation to structural damage, resulting in ankylosis or fusion of vertebrae, significantly impairing quality of life. Current treatment strategies for AS focus on symptom control, reducing inflammation, and preventing structural damage. First-line therapies include nonsteroidal anti-inflammatory drugs (NSAIDs), while biologics, such as TNF inhibitors and IL-17 inhibitors, are reserved for patients with refractory or severe disease [9]. Similar to CD, the management of AS requires a personalized approach, as patients exhibit varying responses to therapy.

The association between CD and AS has attracted significant attention in the medical community, reflecting the shared clinical, genetic, and immunological underpinnings of these conditions [10,11,12]. Epidemiological studies have demonstrated an increased prevalence of AS in patients with CD and vice versa, suggesting a bidirectional relationship [10,13]. This overlap is not coincidental but points to a complex interplay of shared genetic susceptibility loci, immune dysregulation, and environmental triggers [14,15,16]. Among the genetic factors, variations in genes within the major histocompatibility complex (MHC), particularly HLA-B27, have been implicated in AS pathogenesis, while NOD2 and other loci are strongly associated with CD [14,17]. Notably, the IL-23/Th17 axis plays a critical role in the inflammatory cascades of both diseases, providing a mechanistic link and a therapeutic target for biologic agents in AS but not CD [15].

In addition to genetic predisposition, environmental factors such as smoking, diet, and gut microbiome composition have been implicated in the pathogenesis of both CD and AS [18,19]. Dysbiosis, or an imbalance in gut microbial communities, is a common feature of these conditions, leading to a pro-inflammatory milieu that perpetuates systemic and localized inflammation [18]. Such findings underscore the importance of investigating the shared pathways and triggers of CD and AS to better understand their overlapping pathophysiology.

The potential implications of this association extend beyond understanding pathogenesis. Clinically, the coexistence of CD and AS poses unique challenges in disease management. Patients with both conditions may exhibit more severe disease phenotypes, higher rates of treatment refractoriness, and increased healthcare utilization [20,21]. For instance, individuals with CD and concurrent AS may require early initiation of biologic therapies that target shared inflammatory pathways, such as TNF or JAK inhibitors, to achieve adequate disease control [19]. Furthermore, the extra-intestinal manifestations of CD, such as arthropathy, can complicate diagnostic and therapeutic decision-making, necessitating a multidisciplinary approach involving gastroenterologists, rheumatologists, and other specialists [21].

Advances in genetic research have provided novel insights into the association between CD and AS. Among these, two-sample Mendelian randomization (TSMR) has emerged as a powerful analytical approach for elucidating causal relationships in complex diseases (Figure 1) [22,23]. By leveraging genetic variants as instrumental variables, TSMR minimizes confounding and reverse causation, offering a robust framework for exploring the genetic interplay between risk factors and disease outcomes [24]. This methodology has been successfully applied to investigate the shared genetic architecture of various autoimmune diseases, highlighting its potential to uncover novel therapeutic targets and refine personalized medicine approaches [24].

In the context of CD and AS, TSMR offers a unique opportunity to explore whether genetic susceptibility to one condition influences disease severity or treatment outcomes in the other. For example, genetic predisposition to AS may contribute to a more aggressive disease phenotype in CD, necessitating advanced therapeutic interventions [25,26]. Understanding such causal relationships is critical for developing predictive models and tailoring treatment strategies for patients with overlapping features of these conditions.

Despite the growing body of evidence supporting the association between CD and AS, substantial gaps remain in our understanding of their interconnected causal dynamics, particularly regarding disease progression and treatment responses in CD [10,25,26]. Current research has predominantly focused on epidemiological and genetic associations, with limited exploration of how these associations translate into clinical practice [10,25,26]. Addressing these gaps is essential for optimizing patient outcomes and informing evidence-based guidelines for managing these complex conditions.

This study aims to investigate whether a genetic predisposition to AS correlates with an elevated likelihood of requiring advanced therapy in CD, potentially signaling a more severe disease phenotype. By employing the robust analytical framework of TSMR, this research seeks to establish a causal link between AS-related genetic variants and the necessity for intensive treatment regimens in CD. Such findings could have significant implications for clinical practice, highlighting the need for early and aggressive intervention strategies in patients at risk of severe disease progression. Moreover, insights gained from this study could inform the development of personalized treatment plans, leveraging genetic and clinical data to improve disease management and patient outcomes.

## 2. Methods

### 2.1. Data Sources

This study utilized a two-sample Mendelian randomization (TSMR) approach to investigate the causal relationship between genetic susceptibility to Ankylosing Spondylitis (AS) and the requirement for second-line therapy in Crohn’s disease (CD). The exposure data for AS was obtained from the GWAS dataset with ID ebi-a-GCST005529, published by Cortes et al. in 2013 [20]. This dataset included 10,619 participants and 25 genome-wide significant single-nucleotide polymorphisms (SNPs) that served as instrumental variables. The outcome dataset, focused on the need for second-line medication in CD, was sourced from the FinnGen project under the ID “finn-b-RX_CROHN_2NDLINE”. This dataset, published in 2021, contained 16,380,466 SNPs, of which 24 genome-wide significant SNPs were identified as instrumental variables [22]. (Figure 2 presents a flowchart of our TSMR analysis).

### 2.2. Instrumental Variable Selection

Instrumental variables (IVs) were selected from single-nucleotide polymorphisms (SNPs) that reached genome-wide significance (*p* < 5.0 × 10^−8^) [27]. To curtail bias due to linkage disequilibrium (LD), SNPs were clumped with an r^2^ < 0.001 within a 10,000 kb window. The selected IVs from the GWAS datasets for each health outcome were rigorously documented, detailing effect alleles, betas, standard errors, and *p*-values. The strength of the instrumental variables (IVs) used to predict the genetic risk of Ankylosing Spondylitis was quantified using the *F*-statistic, calculated based on the following formula:$$F=\frac{R^{2}(N-1-K)}{(1-R^{2})K}$$

In this formula, *R*^2^ represents the proportion of variance in the exposure that is explained by the IVs, *N* denotes the total number of samples in the analysis, and *K* indicates the number of IVs employed. An F-statistic value of 10 or higher is generally considered sufficient to mitigate the risk of weak instrument bias.

Research Design Assumptions

Our MR analysis was predicated on three critical assumptions:

The IVs are significantly associated with the exposure (AS);

The IVs are not associated with any confounders of the exposure–outcome relationship;

The IVs affect the outcomes exclusively through their impact on the exposure.

### 2.3. Data Harmonization

Data harmonization was conducted to align SNP effect estimates between the AS and CD datasets. This process ensured consistent allelic representation for both exposure and outcome data. SNPs with strand mismatches were aligned to the same reference strand, and palindromic SNPs with ambiguous allele frequencies were excluded to prevent errors. Minor allele frequencies (MAFs) ≤ 0.01 were subjected to additional quality control measures to address potential strand ambiguities [28].

### 2.4. Statistical Analysis

In our MR analysis, we harmonized SNP effects on AS and needed second-line treatment for CD using MRBase, ensuring allele consistency. Our Two-Sample MR approach encompassed multiple methodologies: the primary inverse-variance weighted (IVW) method integrated SNP data to estimate causal effects, while the weighted median, MR Egger, simple mode, and weighted mode analyses provided supplementary insights, including checks for horizontal pleiotropy. Sensitivity and the robustness of our results were vetted through leave-one-out and MR Steiger tests, the latter confirming the temporality of the genetic relationship. We performed MR Egger regression to assess potential pleiotropy and provide unbiased causal estimates even in the presence of pleiotropic genetic variants. Incorporating the MR-PRESSO tool allowed us to detect and adjust for outliers. Power calculations assessed our sample’s sufficiency, outlined by Brion et al. [18] (available on https://shiny.cnsgenomics.com/mRnd/, accessed on 8 April 2024). All statistical procedures were performed via the MRBase web application [19] and R (Version 2023.03.0+386), utilizing the TwoSampleMR and MR-PRESSO packages, using an alpha of 0.05 to define statistical significance.

## 3. Results

This study utilized two-sample Mendelian randomization (TSMR) to examine the potential causal link between genetic predisposition to AS and the need for second-line medication in CD. The exposure variable, AS, was detailed with 25 genetic instruments derived from the genome-wide association study (GWAS) ID ebi-a-GCST005529. This GWAS, authored by Cortes et al. and published in 2013 [20], encompassed a sample size of 22,647 participants. The outcome variable, the need for second-line medication in CD, was based on a separate GWAS with ID finn-b-RX_CROHN_2NDLINE from the FinnGen project from 2021 [22]. This study identified 24 relevant genetic instruments out of a total of 16,380,466 SNPs in the GWAS dataset.

### 3.1. Casual Association Analysis and Sensitivity Analysis

The MR analysis, encompassing various methods, revealed a significant association. Specifically, the inverse-variance weighted (IVW) method indicated an Odds Ratio (OR) of 2.16 (95% CI: 1.59 to 2.94, *p* = 9.120082 × 10^−8^), and the MR Egger method showed an OR of 2.71 (95% CI: 1.58 to 4.62, *p* = 0.000560968). These findings were corroborated by the weighted median, simple mode, and weighted mode methods, all indicating statistical significance (Table 1 and Figure 3 and Figure 4).

The leave-one-out analysis highlighted the influence of SNP rs2517655 on the results (Figure 5). Its exclusion notably affected the findings, yet they remained statistically significant. Post-exclusion, the IVW method showed an OR of 1.90 (95% CI: 1.43 to 2.53, *p* = 3.328640 × 10^−6^), and the MR Egger method showed an OR of 2.23 (95% CI: 1.42 to 3.51, *p* = 0.002853446). The other methods, namely the weighted median, simple mode, and weighted mode, continued to support these findings, confirming the robustness and significance of the results even after the exclusion of this influential SNP (Table 1).

### 3.2. Heterogeneity and Pleiotropy Tests

Significant heterogeneity in the results was indicated (Q = 91.33226, Q_df = 22, and Q_pval = 2.031352 × 10^−10^ for MR Egger; Q = 96.61781, Q_df = 23, and Q_pval = 5.369275 × 10^−11^ for IVW). The MR Egger regression for pleiotropy showed an intercept of −0.01411563 (SE = 0.01250996, *p* = 0.2713267), suggesting minimal evidence of pleiotropy. The forest plot analysis demonstrated a concentration of IVW estimates around the null effect line, contrasting with the more dispersed MR Egger estimates, indicating potential instrument pleiotropy (Table 2, Figure 6).

### 3.3. MR-PRESSO Results

MR-PRESSO identified potential outliers, with significant distortion indicated in the global test (RSSobs = 106.0074, *p* value = “<0.001”). Outlier correction adjusted the causal estimate to OR = 2.04 (95% CI: 1.67 to 2.50), *p*-value = 9.976886 × 10^−7^, with three SNPs identified as outliers.

### 3.4. MR Egger Regression and Two-Stage Least Squares

The MR Egger regression provided an overall OR of 2.43 (95% CI: 1.73 to 3.41, *p* = 2.127124 × 10^−5^). The two-stage least squares analysis indicated strong instrument strength (F-statistic= 98.67) and association power (YZ association NCP = 57.53, Power = 1.00). (Table 2, Figure 7).

## 4. Discussion

In our investigation, we applied TSMR to assess the genetic predisposition to AS and its causal influence on the requirement for second-line medication in CD. Our findings from multiple MR methodologies consistently demonstrated a significant association: the primary IVW analysis yielded an OR of 2.16, while MR Egger presented an OR of 2.71, both with 95% confidence intervals indicating robust statistical significance. Notably, the exclusion of SNP rs2517655—identified as an influential outlier associated with rheumatoid arthritis according to the SCAN database [24]—substantially altered the results; however, the association remained significant. This underscores the potential genetic interplay between AS and advanced therapeutic needs in CD. These results suggest that genetic factors contributing to AS may indeed increase the likelihood of progressing to second-line medication in CD, highlighting the importance of genetic screening in the clinical management of these patients.

Current research indicates a complex relationship between CD and AS, suggesting shared genetic and pathophysiological factors [29]. A recent study highlighted the role of gut inflammation in predicting the development of CD in patients with existing AS [25]. Elevated levels of fecal calprotectin, a marker of gut inflammation, were found to be a strong predictor of CD development within five years in patients with AS [29]. Baseline levels of fecal calprotectin were notably higher in AS patients who were later diagnosed with CD compared to those who did not develop CD [25]. These findings emphasize the potential utility of gut inflammation markers in monitoring and predicting disease progression in at-risk populations.

Further research supports the genetic link between AS and CD [30]. A genome-wide association study found that several susceptibility genes for CD are also associated with AS [30]. This association was observed even in AS cases without clinically evident CD. Proteins encoded by STAT3, IL12B, and IL23R, which are essential for producing interleukin-17 by type 17 T helper cells, play a significant role in the pathogenesis of both AS and CD [30].

Additionally, AS and CD frequently occur in the same families and individuals, with approximately 10% of AS patients presenting with overt IBD and 50–70% showing subclinical inflammation in the terminal ileum [31]. The presence of the HLA-B27 gene is strongly associated with both AS and CD, particularly in patients with spondylitis/sacroiliitis [31,32]. This overlap is further supported by epidemiological evidence showing a bidirectional relationship, where patients with CD are at an increased risk of developing AS and vice versa [29,33].

The complex connection between IBD and AS has long been the subject of extensive research, acknowledging their common occurrence and hinting at overlapping genetic and inflammatory mechanisms [10,20,34]. The hypothesis that an inherent increased risk of IBD might also elevate the risk of AS, as suggested by previous Mendelian randomization studies, provides a framework for interpreting our findings [35]. These earlier studies suggest that genetic susceptibility to AS may exacerbate inflammatory pathways in CD, leading to a more severe disease phenotype. This observation aligns with the research by Savin et al., which highlighted the necessity for more advanced therapeutic approaches in patients with concurrent AS and CD [26].

Increased therapeutic requirements in such patients could stem from the augmented inflammatory burden associated with shared pathways, such as those involving IL-23 and IL-17. Therapies targeting shared pathways, including TNF inhibitors and Janus kinase (JAK) inhibitors (e.g., upadacitinib), offer promising options for managing CD in patients with a genetic predisposition to AS [36]. These treatments have demonstrated efficacy in reducing inflammation in both diseases and may serve as key strategies for addressing refractory cases. However, their use in clinical practice requires further validation through randomized controlled trials and expert consensus to ensure optimal outcomes [37].

The joint–gut axis hypothesis offers additional insights into the interplay between AS and CD. This concept posits that shared genetic vulnerabilities, combined with environmental and host factors, can trigger a systemic inflammatory response affecting both gut and joint tissues [29,35]. Activated gut-resident immune cells, such as Th17 cells and macrophages, may migrate to the joints, perpetuating inflammation in genetically predisposed individuals [31]. This bidirectional interaction between gut and joint inflammation highlights the importance of addressing both components in therapeutic strategies.

Microbiota-driven inflammation has also emerged as a critical factor linking AS and CD. Research identified significant differences in the gut microbiota of patients with spondyloarthritis, including those with subclinical gut inflammation [38]. Dysbiosis in these patients is characterized by reduced microbial diversity and an overrepresentation of pro-inflammatory taxa. This altered microbial composition is thought to disrupt intestinal barrier integrity and promote systemic inflammation, potentially exacerbating disease severity. Such findings underscore the potential of microbiota-targeted therapies, including probiotics, prebiotics, and fecal microbiota transplantation (FMT), as adjunctive treatments in patients with overlapping AS and CD features.

The convergence of inflammatory pathways in AS and CD, particularly the Th17 axis, further underscores the interconnected nature of these diseases. Recent studies have highlighted the efficacy of biologics targeting these pathways in reducing inflammation and achieving disease remission [26,34]. For example, JAK inhibitors, which modulate multiple inflammatory pathways, offer an alternative for patients who do not respond to traditional biologics [36].

Another key aspect of this relationship is the role of genetic markers in predicting disease severity and treatment response. CARD15 gene polymorphisms, for instance, have been associated with chronic gut inflammation in spondyloarthritis patients [8]. These polymorphisms may serve as genetic markers for identifying patients at risk for more severe disease phenotypes, such as CD-related sacroiliitis. This insight could inform personalized treatment strategies, enabling clinicians to tailor interventions based on individual genetic profiles.

The clinical implications of these findings are profound. Patients with coexisting AS and CD often require multidisciplinary management, involving gastroenterologists, rheumatologists, and other specialists. Early genetic screening for shared susceptibility loci, such as HLA-B27 and CARD15, could facilitate the identification of at-risk individuals and enable proactive treatment planning [8]. Additionally, incorporating gut inflammation markers, such as fecal calprotectin, into routine monitoring protocols could enhance the early detection of disease progression in AS patients, allowing for timely therapeutic intervention.

Our findings also emphasize the importance of addressing extraintestinal manifestations in CD patients with concurrent AS. Joint involvement, including sacroiliitis and peripheral arthritis, can complicate disease management and contribute to treatment refractoriness. Biologic therapies targeting shared inflammatory pathways may offer dual benefits in controlling both intestinal and joint inflammation, reducing the overall disease burden and improving quality of life.

Emerging evidence further suggests that environmental factors, such as diet, smoking, and gut microbiota composition, may modulate the genetic predisposition to AS and CD [18,19]. Smoking, for example, has been shown to exacerbate inflammation in both diseases, highlighting the need for comprehensive lifestyle interventions alongside pharmacological treatments. Similarly, dietary modifications aimed at reducing pro-inflammatory triggers could complement existing therapies and enhance patient outcomes.

Despite significant advances in understanding the genetic and immunological underpinnings of AS and CD, many questions remain unanswered. Future research should aim to elucidate the precise mechanisms by which shared genetic factors influence disease progression and treatment response. Large-scale, multi-ethnic studies are needed to validate our findings and explore their applicability across diverse populations. Additionally, integrating genomic data with clinical and environmental variables could provide a more comprehensive understanding of these complex diseases, paving the way for truly personalized medicine.

The intricate relationship between AS and CD exemplifies the challenges of managing immune-mediated diseases with overlapping pathophysiology. Our study contributes to the growing body of evidence highlighting the genetic and inflammatory connections between these conditions. By advancing our understanding of these mechanisms, we can develop more effective strategies for managing patients with coexisting AS and CD, ultimately improving their quality of life and long-term outcomes.

The efficacy of anti-TNF therapies, particularly infliximab and adalimumab, is reaffirmed by our findings as these agents are crucial for patients with concurrent IBD and spondyloarthritis, particularly when standard treatments fail [39,40]. This correlation is particularly relevant given that anti-TNF agents are generally employed as advanced treatments for Crohn’s disease [26,40], aligning with our results which suggest a genetic predisposition in AS patients towards a higher necessity for advanced therapeutic interventions in CD. Our study’s genetic insights into AS may thus influence the therapeutic strategy, especially regarding the earlier initiation of anti-TNF therapy in CD patients with concomitant AS. Importantly, these insights pave the way for genetically tailored treatments, advancing personalized medicine for patients with these complex immune-mediated diseases.

### 4.1. Strengths and Limitations of This Study

One of the principal strengths of our study lies in the application of two-sample Mendelian randomization (TSMR), a robust method for inferring causal relationships, by addressing confounding and reverse causation, common challenges in observational research [16]. By leveraging extensive genome-wide association study (GWAS) datasets, we utilized multiple Mendelian randomization (MR) methods, including inverse-variance weighting (IVW), MR Egger, and the weighted median approach, to ensure the reliability of our findings. The robustness of the results was further validated through comprehensive sensitivity analyses and the MR regression test, which assessed the presence of horizontal pleiotropy [41].

Nonetheless, our study is not without limitations. First, the predominant use of European ancestry data in the GWAS datasets limits the generalizability of the findings to other populations, highlighting the need for more diverse genetic studies [42]. Second, despite our efforts to minimize bias, the possibility of residual pleiotropy cannot be completely ruled out. Third, while genetic associations provide valuable insights, they do not fully capture other critical factors, such as environmental or lifestyle influences, which also contribute to the pathogenesis of these diseases.

Additionally, although stringent harmonization procedures were employed to align the datasets, potential limitations such as residual strand alignment errors or the exclusion of informative single-nucleotide polymorphisms (SNPs) could introduce minor inconsistencies. Nevertheless, sensitivity analyses demonstrated that our main results were robust, suggesting that harmonization artifacts had minimal impact on this study’s conclusions. Future research should address these limitations to strengthen the applicability of our findings.

### 4.2. Conclusion and Future Research Directions

In summary, our study elucidates a crucial genetic connection between Ankylosing Spondylitis (AS) and the increased need for second-line treatments in Crohn’s disease (CD), underscoring the role of genetic predisposition in determining treatment pathways, and maybe suggesting the possibility of a more severe form of CD in patients with AS, characterized by a different disease progression. This finding advocates for early screening in patients with co-existing AS and CD, facilitating the modification of treatment strategies at an earlier stage. Such an approach promises to enhance patient outcomes by tailoring interventions more closely to individual genetic profiles and using more tailored therapies for the shared inflammatory pathways like JAK inhibitors. This insight into the genetic interplay between AS and CD not only enhances our understanding of these complex conditions but also opens avenues for more personalized and effective medical interventions in the management of inflammatory diseases.

## Figures and Tables

**Figure 1 jcm-13-07496-f001:**
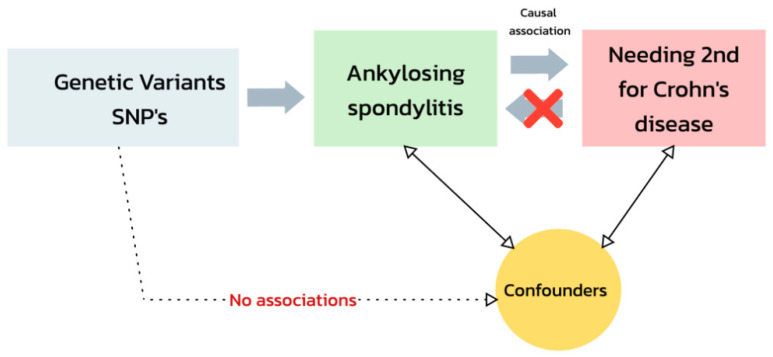
A summarized illustration of the mendelian randomization causal association assessment between Ankylosing Spondylitis (exposure) and the need for a second-line therapy in Crohn’s disease (outcome).

**Figure 2 jcm-13-07496-f002:**
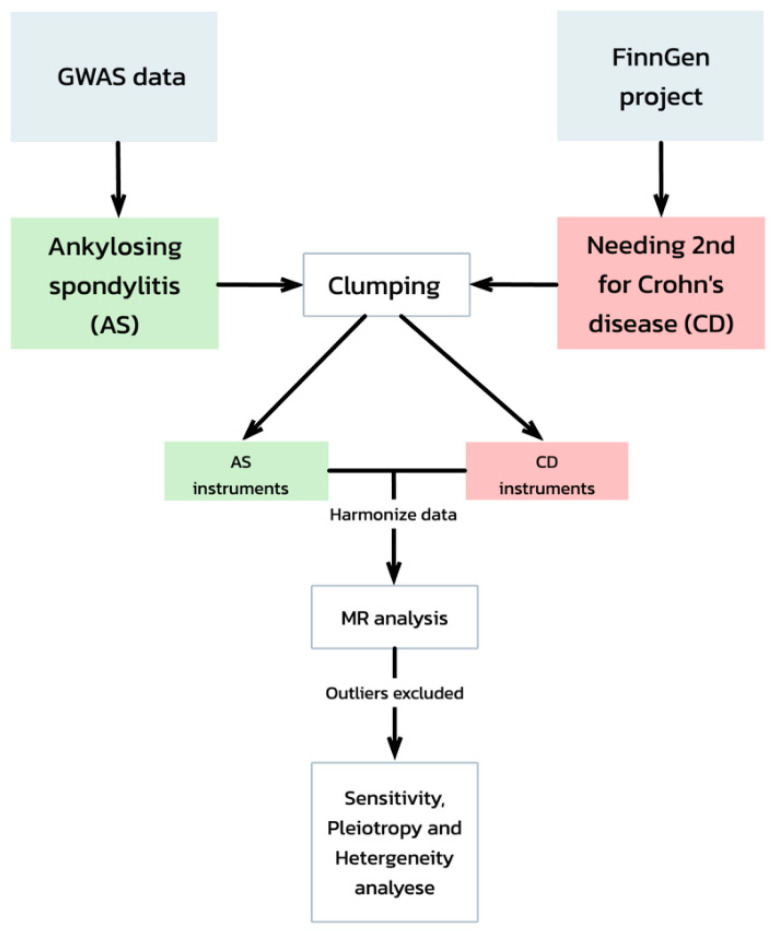
A comprehensive flowchart of our Mendelian randomization analysis.

**Figure 3 jcm-13-07496-f003:**
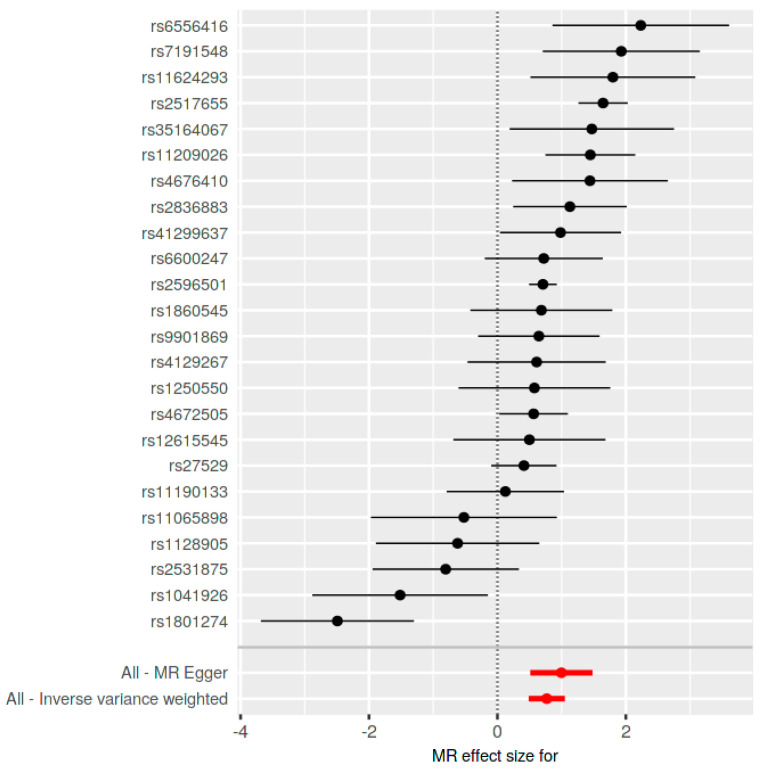
Forest plot of SNP effects on Ankylosing Spondylitis and the need of second-line medication for Crohn’s disease.

**Figure 4 jcm-13-07496-f004:**
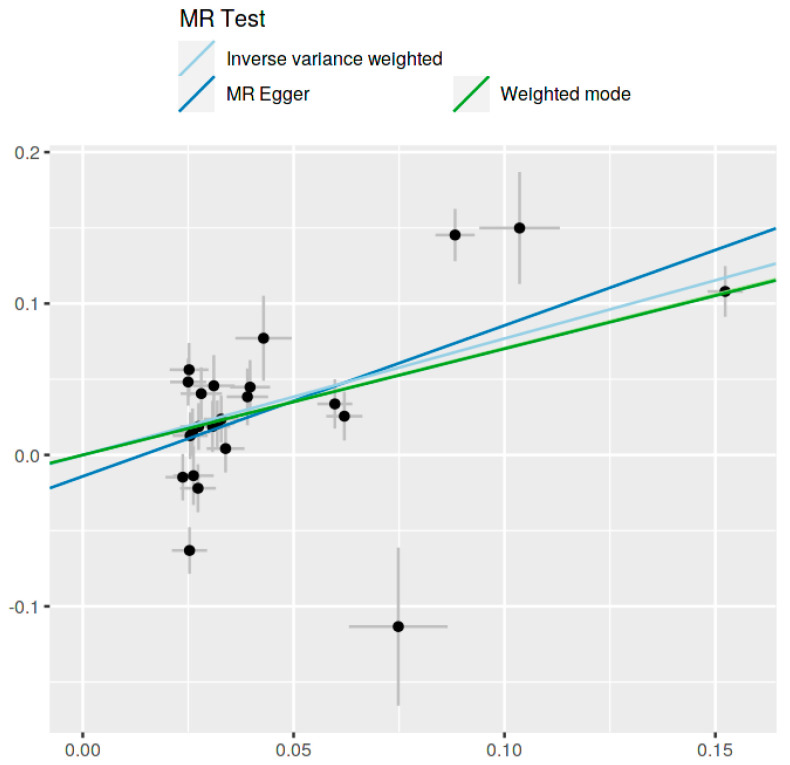
Scatter plot demonstrating the relationship between SNP effects on Ankylosing Spondylitis and the need of second-line medication for Crohn’s disease across multiple MR methods.

**Figure 5 jcm-13-07496-f005:**
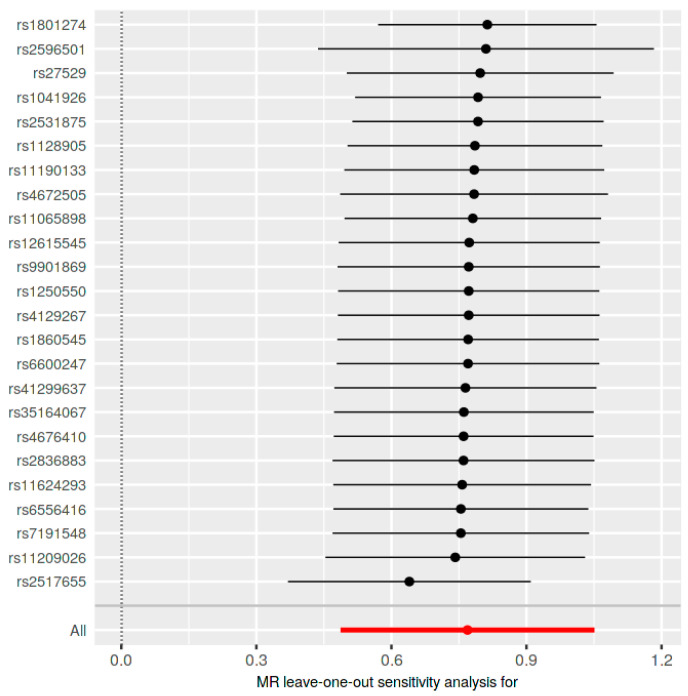
Leave-one-out sensitivity analysis.

**Figure 6 jcm-13-07496-f006:**
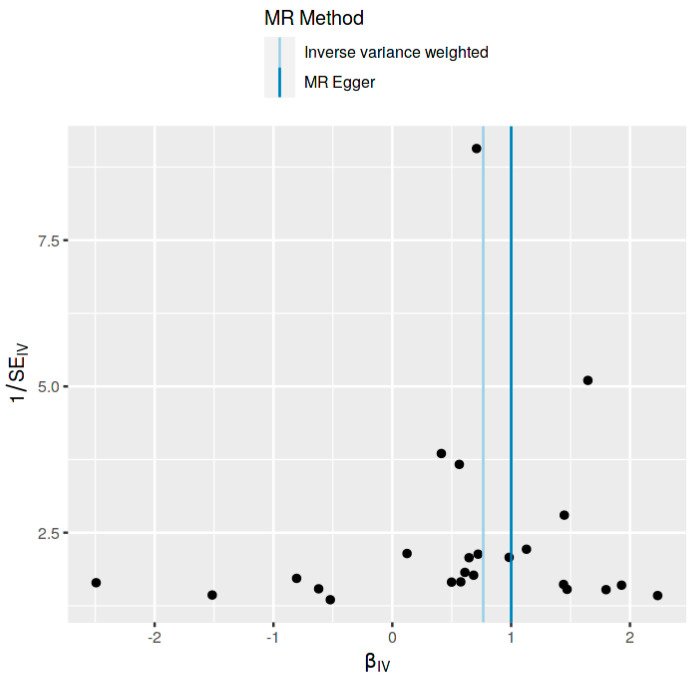
Funnel plot assessing the dispersion and symmetry of SNP effects in MR analysis.

**Figure 7 jcm-13-07496-f007:**
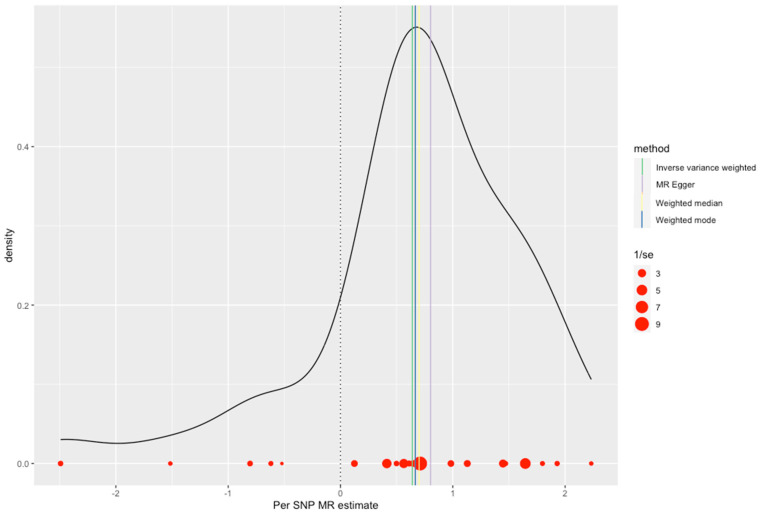
Density plot and individual SNP effects highlighting the distribution and precision of estimates in MR analysis.

**Table 1 jcm-13-07496-t001:** A summary of the casual estimates in the different MR methods, before and after the exclusion of SNP rs2517655.

MR Method	Status	OR (95% CI)	*p*-Value
Inverse-Variance Weighted (IVW)	Before Exclusion	2.16 (1.59 to 2.94)	9.120082 × 10^−8^
	After Exclusion	1.90 (1.43 to 2.53)	3.328640 × 10^−6^
MR Egger	Before Exclusion	2.71 (1.58 to 4.62)	0.000560968
	After Exclusion	2.23 (1.42 to 3.51)	0.002853446
Weighted Median	Before Exclusion	2.03 (1.65 to 2.49)	1.265734 × 10^−11^
	After Exclusion	2.01 (1.64 to 2.47)	2.207916 × 10^−11^
Simple Mode	Before Exclusion	2.11 (1.33 to 3.34)	0.00354507
	After Exclusion	1.95 (1.29 to 2.95)	0.005307088
Weighted Mode	Before Exclusion	2.02 (1.65 to 2.53)	7.888539 × 10^−7^
	After Exclusion	1.95 (1.29 to 2.95)	2.064498 × 10^−6^

**Table 2 jcm-13-07496-t002:** Summary of Mendelian randomization analyses.

Analysis	Heterogeneity Test (MR Egger; IVW)	Pleiotropy Test	MR-PRESSO (Global Test; Outliers)	MR Egger Regression (Intercept)
Results	*p* = 2.03 × 10^−10^; *p* = 5.37 × 10^−11^	*p* = 0.2713	*p* < 0.001; Outliers: 1, 9, 11	*p* = 0.3455

## Data Availability

The original contributions presented in the study are included in the article, further inquiries can be directed to the corresponding author.

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
