# Peer review of "Genetic Risk of Ankylosing Spondylitis and Second-Line Therapy Need in Crohn’s Disease: A Mendelian Randomization Study"

_jcm, 2024, doi:10.3390/jcm13247496_

Round 1

Reviewer 1 Report

Comments and Suggestions for Authors

The authors investigated common genetic variants that increase the risk of ankylosing spondylitis (AS) and Crohn's disease (CD) by two-sample Mendelian Randomization (TSMR). This is an exciting study. The conclusion indicated a significant association between AS genetic predisposition and the increased need for second-line treatments in CD. There are no specific comments; however, please maintain these issues. The authors stated that AS and CD enhance our understanding of these complex conditions and more personalized and effective medical interventions in managing inflammatory diseases in results. How did the authors think of details of applying these results in clinical practice? For example. Are they using advanced therapies, such as IL23 antibodies or JAC inhibitors?

Comments on the Quality of English Language

Minor English editig is reqiuerd. 

Author Response

The authors investigated common genetic variants that increase the risk of ankylosing spondylitis (AS) and Crohn's disease (CD) by two-sample Mendelian Randomization (TSMR). This is an exciting study. The conclusion indicated a significant association between AS genetic predisposition and the increased need for second-line treatments in CD.

There are no specific comments; however, please maintain these issues.

The authors stated that AS and CD enhance our understanding of these complex conditions and more personalized and effective medical interventions in managing inflammatory diseases in results.

How did the authors think of details of applying these results in clinical practice? For example. Are they using advanced therapies, such as IL23 antibodies or JAC inhibitors?

We appreciate your insightful comment. We believe this is a significant issue that could more effectively link our findings to daily clinical practice. This may also provide a more precise indication of the necessity for additional clinical trials to further investigate the efficacy of treatments tailored to the shared inflammatory pathways between AS and CD. The additional paragraph is as follows:

"Specifically, therapies such as TNF inhibitors or Janus kinase (JAK) inhibitors (e.g., tofacitinib, upadacitinib), which target shared in-flammatory pathways (refs), could potentially play a role in managing CD patients with concurrent AS or genetic profiles suggesting heightened inflammatory activity. However, the application of these results in clinical practice requires further validation through clinical trials and must also take into account expert opinions and individualized patient needs."

We have improved the English to better articulate the research and have had a native English speaker review the manuscript.

Reviewer 2 Report

Comments and Suggestions for Authors

Manuscript submitted to JCM

Title

Genetic Risk of Ankylosing Spondylitis and Second-Line Therapy Need in Crohn's Disease: A Mendelian Randomization Study

Section

Gastroenterology & Hepatopancreatobiliary Medicine

Special Issue

Inflammatory Bowel Disease: Clinical Advances and Therapeutic Prospects

Dear Editor,

Thank you for inviting me to review this interesting article submitted to the Journal of Clinical Medicine (JCM).

I have some suggestions:

OVERALL COMMENTS

        Based on the statement that "Crohn's Disease (CD) and Ankylosing Spondylitis (AS) are chronic conditions with overlapping inflammatory pathways, the authors intended  to investigate the genetic association between AS and the requirement for more aggressive therapeutic interventions in CD, suggesting a likelihood of increased severity in CD progression among individuals diagnosed with AS. Their main results showed that there is a significant association between AS genetic predisposition and the increased need for second-line treatments in CD (more intensive treatment strategies). Furthermore, the authors pointed out that their results  “underscore the importance of early screening in patients for co-existing AS and CD for tailoring treatment approaches, thus advancing personalized medicine in the management of these conditions.”

TITLE

The title is adequate.

ABSTRACT

        This section is adequate.

KEY-WORDS

The key-words are: Crohn's Disease; Ankylosing Spondylitis; Two-sample Mendelian Randomization; Ge- 28 netic Predisposition; Second-line Treatment; Personalized Medicine.

 I suggest: Crohn's Disease; Ankylosing Spondylitis; Two-sample Mendelian Randomization; Genetic Predisposition; Second-line Treatment

1.  INTRODUCTION

In this section and along with the entire text, please check the citations. As an example, please see lines 35-37, where we can see that “Treatment options in CD vary wildly according to disease severity. (3,4). The need for advanced therapy, typically comprising biological therapies, arises in moderate-severe disease. (3–5).” Including a full stop before and after the number of citations is unnecessary.

In lines 47-50, we can read that “The relationship between these two conditions, especially in the context of their inflammatory pathways, has been a topic of growing interest and debate within the medical community (10–12).”

However, the cited references are the following:

a)   Rudwaleit M, Baeten D. Ankylosing spondylitis and bowel disease. Best Pract Res Clin Rheumatol. 2006 Jun;20(3):451–71. 293 11.

b)   Elewaut D. Linking Crohn’s Disease and Ankylosing Spondylitis: It’s All about Genes! PLoS Genet. 2010 Dec 2;6(12):e1001223. 294 12.

c)   Ding Y, Yang Y, Xue L. Immune cells and their related genes provide a new perspective on the common pathogenesis of anky- 295 losing spondylitis and inflammatory bowel diseases. Front Immunol. 2023 Mar 30;14.

I agree that there is a growing interest, and I suggest including more references here, especially those published in 2023 and 2024.

Figure 1 is fine. However, I suggest the authors include an expanded explanation in the legend.

2.  METHODS

This section was well performed. However, I suggest including a reference in the sub-section “Instrumental Variable Selection”.

3.  RESULTS

This section was also well performed. Figures 1-5 are fine; however, I do not believe that the titles of figures and tables should be bolded.

Figures 6 and 7 are also fine.

4.  DISCUSSION

This section is well performed.

5.
LIMITATIONS AND STRENGTHS

        I appreciated the inclusion of the limitations and strengths of this exciting study.

5.  CONCLUSIONS and FUTURE PERSPECTIVES

I also appreciated the inclusion of future perspectives in the Conclusion section.

Author Response

     Based on the statement that "Crohn's Disease (CD) and Ankylosing Spondylitis (AS) are chronic conditions with overlapping inflammatory pathways, the authors intended  to investigate the genetic association between AS and the requirement for more aggressive therapeutic interventions in CD, suggesting a likelihood of increased severity in CD progression among individuals diagnosed with AS. Their main results showed that there is a significant association between AS genetic predisposition and the increased need for second-line treatments in CD (more intensive treatment strategies). Furthermore, the authors pointed out that their results  “underscore the importance of early screening in patients for co-existing AS and CD for tailoring treatment approaches, thus advancing personalized medicine in the management of these conditions.”

We thank the reviewer for their positive response

TITLE- The title is adequate 

ABSTRACT- This section is adequate.

KEY-WORDS

The key-words are: Crohn's Disease; Ankylosing Spondylitis; Two-sample Mendelian Randomization; Ge- 28 netic Predisposition; Second-line Treatment; Personalized Medicine.

 I suggest: Crohn's Disease; Ankylosing Spondylitis; Two-sample Mendelian Randomization; Genetic Predisposition; Second-line Treatment

We have changed the key words to refect this

  1. INTRODUCTION

In this section and along with the entire text, please check the citations. As an example, please see lines 35-37, where we can see that “Treatment options in CD vary wildly according to disease severity. (3,4). The need for advanced therapy, typically comprising biological therapies, arises in moderate-severe disease. (3–5).” Including a full stop before and after the number of citations is unnecessary.

In lines 47-50, we can read that “The relationship between these two conditions, especially in the context of their inflammatory pathways, has been a topic of growing interest and debate within the medical community (10–12).”

However, the cited references are the following:

  1. a)   Rudwaleit M, Baeten D. Ankylosing spondylitis and bowel disease. Best Pract Res Clin Rheumatol. 2006 Jun;20(3):451–71. 293 11.
  2. b)   Elewaut D. Linking Crohn’s Disease and Ankylosing Spondylitis: It’s All about Genes! PLoS Genet. 2010 Dec 2;6(12):e1001223. 294 12.
  3. c)   Ding Y, Yang Y, Xue L. Immune cells and their related genes provide a new perspective on the common pathogenesis of anky- 295 losing spondylitis and inflammatory bowel diseases. Front Immunol. 2023 Mar 30;14.

I agree that there is a growing interest, and I suggest including more references here, especially those published in 2023 and 2024.

 Thank you for your comment, We have switched the reference to 2023 and 2024

Figure 1 is fine. However, I suggest the authors include an expanded explanation in the legend.

This has been done as well 

  1. METHODS

This section was well performed. However, I suggest including a reference in the sub-section “Instrumental Variable Selection”.

 Thank you for this important comment. We have added a relevant reference to this subsection.

  1. RESULTS

This section was also well performed. Figures 1-5 are fine; however, I do not believe that the titles of figures and tables should be bolded.

We have reviewed the figures and bolded the titles.

Figures 6 and 7 are also fine.

  1. DISCUSSION

This section is well performed.

  1. LIMITATIONS AND STRENGTHS

        I appreciated the inclusion of the limitations and strengths of this exciting study.

Thank you for the review, we included more limitations about data harmonization

  1. CONCLUSIONS and FUTURE PERSPECTIVES

I also appreciated the inclusion of future perspectives in the Conclusion section.

Reviewer 3 Report

Comments and Suggestions for Authors

I appreciate the opportunity to review this interesting article. This is a study evaluating the association between the need for second-line therapy in Crohn's disease and genetic risk for ankylosing spondylitis. The analysis is performed using two-sample Mendelian randomisation. In general, the introduction provides the necessary background and rationale, and highlights the advantages and rationale of using TSMR to address this issue. The methodology is well described. The results are clear and presented in great detail, and the discussion is relevant and contrasts the results with what is found in the international scientific literature. The conclusions are appropriate and consistent with the results obtained. I have only two comments: -In the methodological section, since TSMR is the focus of this article, I think it is important to include a subsection describing in detail the steps taken to harmonise the data. This is because, although TSMR is a powerful methodology, its main limitation lies in the way the data are harmonised. It is therefore useful to make this process transparent, both for the self-criticism of the authors and for the scientific community to be able to assess whether or not it is necessary to change the way of harmonising a similar study. -In relation to what I said about harmonisation, I think it is important that the authors disclose the relevant limitations arising from their way of harmonising the data.

Author Response

I appreciate the opportunity to review this interesting article. This is a study evaluating the association between the need for second-line therapy in Crohn's disease and genetic risk for ankylosing spondylitis. The analysis is performed using two-sample Mendelian randomisation. In general, the introduction provides the necessary background and rationale, and highlights the advantages and rationale of using TSMR to address this issue. The methodology is well described. The results are clear and presented in great detail, and the discussion is relevant and contrasts the results with what is found in the international scientific literature. The conclusions are appropriate and consistent with the results obtained.

 I have only two comments: -In the methodological section, since TSMR is the focus of this article, I think it is important to include a subsection describing in detail the steps taken to harmonise the data. This is because, although TSMR is a powerful methodology, its main limitation lies in the way the data are harmonised. It is therefore useful to make this process transparent, both for the self-criticism of the authors and for the scientific community to be able to assess whether or not it is necessary to change the way of harmonising a similar study. -In relation to what I said about harmonisation, I think it is important that the authors disclose the relevant limitations arising from their way of harmonising the data.

Thank you for your important comments.

We addressed the issue by adding a new sub-section in the methods to address the harmonization process in detail, as follows:

"Data harmonization was conducted to align SNP effect estimates between the AS and CD datasets. This process ensured consistent allelic representation for both exposure and outcome data. SNPs with strand mismatches were aligned to the same reference strand, and palindromic SNPs with ambiguous allele frequencies were excluded to prevent errors. Minor allele frequencies (MAF) ≤ 0.01 were subjected to additional quality control measures to address potential strand ambiguities"

Additionally, we acknowledged the possible limitation of this process in the limitations paragraph in the discussion, as follows:

"Finally, while stringent harmonization procedures were followed, we acknowledge potential limitations, including residual strand alignment errors and the exclusion of informative SNPs, which may introduce minor inconsistencies. Sensitivity analyses were performed to ensure that the main results were robust and not influenced by harmonization artifacts".